# Optimizing *Levilactobacillus brevis* NPS-QW 145 Fermentation for Gamma-Aminobutyric Acid (GABA) Production in Soybean Sprout Yogurt-like Product

**DOI:** 10.3390/foods12050977

**Published:** 2023-02-25

**Authors:** Yue Zhang, Mengjiao Zhu, Wenjing Lu, Cen Zhang, Di Chen, Nagendra P. Shah, Chaogeng Xiao

**Affiliations:** 1Institute of Food Science, State Key Laboratory for Managing Biotic and Chemical Threats to the Quality and Safety of Agro-Products, Zhejiang Academy of Agricultural Sciences, 298 Desheng Road, Hangzhou 310021, China; 2School of Biological Science, The University of Hong Kong, Pokfulam Road, Hong Kong

**Keywords:** gamma-aminobutyric acid (GABA), fermented soybean sprout yogurt-like product, reversed-phase high performance liquid chromatography, GABA-rich yogurt, response surface methodology

## Abstract

Gamma-aminobutyric acid (GABA) is a non-protein amino acid with various physiological functions. *Levilactobacillus brevis* NPS-QW 145 strains active in GABA catabolism and anabolism can be used as a microbial platform for GABA production. Soybean sprouts can be treated as a fermentation substrate for making functional products. This study demonstrated the benefits of using soybean sprouts as a medium to produce GABA by *Levilactobacillus brevis* NPS-QW 145 when monosodium glutamate (MSG) is the substrate. Based on this method, a GABA yield of up to 2.302 g L^−1^ was obtained with a soybean germination time of one day and fermentation of 48 h with bacteria using 10 g L^−1^ glucose according to the response surface methodology. Research revealed a powerful technique for producing GABA by fermentation with *Levilactobacillus brevis* NPS-QW 145 in foods and is expected to be widely used as a nutritional supplement for consumers.

## 1. Introduction

Gamma-aminobutyric acid (GABA), a four-carbon non-protein and water-soluble amino acid, is the main inhibitory neurotransmitter of the central nervous system [1,2,3,4,5]. It can have beneficial effects on human health and other animals by reducing blood pressure, preventing chronic alcoholic diseases, inhibiting cancer cell proliferation, improving brain function, and promoting insulin [6,7,8]. GABA also demonstrates the potential for lowing blood pressure in spontaneously hypertensive rats (SHR) and hypertensive humans [9,10]. Furthermore, a previous study reported the key role of GABA production in hepatocytes in the dysregulation of glucose regulation and eating behavior associated with obesity [11,12,13]. There has been an increased demand for GABA due to its widespread use in various industries [14].

Concentration of GABA in plant tissues varies between 0.03 and 2.00 μmol g^−1^, increasing with hypoxia, hydraulic pressure, salt stress, temperature shock, germination, and other biotic stresses [4]. Several microorganisms, including lactic acid bacteria (LAB), such as *Levilactobacillus brevis*, *Lacticaseibacillus paracasei*, and *Enterococcus raffinosus*, have recently been intensively investigated and used in GABA synthesis [15], because they are rich in glutamate decarboxylase and can synthesize GABA. 

Plant seed germination is a physiological process that stimulates endogenous enzyme activity and alters biochemical processes [8,16]. According to recent research, soybean sprouts can be utilized as an alternate method to strengthen the nutritional quality of phytochemical content, particularly GABA [7]. Germination of soybean for human consumption would reduce the content of anti-nutritional elements while increasing the number of minerals and phytochemicals such as vitamin E and isoflavone aglycone derivatives [17,18]. In particular, during soybean germination, various free amino acids are produced with protein degradation, providing a natural substrate for GABA synthesis [17]. 

This study aims to use response surface optimization to investigate the effect of soybean germination treatment and lactic acid bacteria fermentation on the level of GABA in soy milk. The study’s results will provide a favorable theoretical basis for producing products with higher nutritional value.

## 2. Materials and Methods

### 2.1. Materials and Strain

Organic soybeans were purchased from a local supplier. Analytical grade chemical reagents utilized in this work were purchased from Sigma-Aldrich Corp., St. Louis, Missouri, USA. *Levilactobacillus brevis* NPS-QW 145 was obtained from BD Company (Franklin Lakes, NJ, USA). Six carbon sources, including glucose, lactose, mannose, malactose, amylopectin, and fructose, were purchased from Sigma-Aldrich Corp., St. Louis, MO, USA. Difeo TM lactobacilli MRS broth and Monosodium glutamate (MSG) were purchased from Difco. (Sparks, MD, USA). All other reagents were of analytical grade.

### 2.2. Preparation of Soybean Sprouts Milk

Germination conditions used in this study were based on Luo’s method [4]. Typically, 200 g of soybeans were selected, washed, and soaked in a 95% ethanol solution for 1 min to remove microorganisms on the surface of soybean seeds. The beans were washed with sterile water and placed in an incubator for germination. Subsequently, the germination status of the beans was observed daily, with the germination length measured as well. Consequently, the bean sprouts were taken out from the incubator on days 0, 1, 3, 4, and 5 to prepare soybean sprout milk. The sprouts were rinsed with clean water and mixed with water in a ratio of 1:2 (soybean sprout: water) before putting the mixture into a grinder for 5 min of pulp grinding. The mixture was then allowed to be filtered, homogenized, and sterilized at 90 °C in a water bath pot before 1 h of boiling. The sterilized mixture was left at room temperature for cooling before fermentation. 

### 2.3. Preparation of Fermented Yogurt-like Product

The fermentation method was conducted following the instructions of Xiao and Shah [19] with slight modifications. Firstly, the soybean sprout yogurt-like product made with sprouts of different germination times was autoclaved and then inoculated with 3% *Lb. brevis* 145 (*v*/*v*), 5 g L^−1^ MSG, and six different monosaccharides (glucose, lactose, mannose, galactose, amylopectin, and fructose) at different concentrations (0, 5, 10, 15, and 20 g L^−1^) and mixed well. Subsequently, the mixture was fermented in the incubator at 37 °C to observe the coagulation state and compare the GABA concentration in it. 

Soybean without germination treatment was used as the blank test. Yogurt prepared from the same quality of milk powder was also fermented with 3% *Lb. brevis* 145 (*v*/*v*), 5 g L^−1^ MSG, and 10 g L^−1^ glucose, and then fermented at 37 °C for 48 h. Moreover, the GABA content in the soybean with germination treatment was compared to explore the effect of germination treatment on the GABA content in the soybean.

### 2.4. Optimization of Fermentation Conditions for the Production of GABA by Lb. brevis 145 in Soybean Sprout Yogurt-like Product

#### 2.4.1. Single-Factor Experiments

*Levilactobacillus brevis* NPS-QW 145 was used as the fermentation strain in a single-factor experiment. The following factors were examined for their influence on the GABA content of fermented soybean sprouts: types of carbon sources (glucose, lactose, mannose, galactose, amylopectin, and fructose), germination time (0, 1, 3, 4, and 5 d), glucose concentration (0, 5, 10, 15, and 20 g L^−1^), and fermentation time (12, 24, 48, 72, and 96 h). The GABA concentration was determined using RP-HPLC (Shimadzu model LC-2010A, Shimadzu Corp., Kyoto, Japan).

#### 2.4.2. Response Surface Methodology (RSM)

RSM is typically used to investigate optimal experimental conditions since it is a reliable and useful statistical methodology. This experimental method was partially modified according to the method of Zhang et al. [14]. Based on the results of the single-factor experiments, glucose concentration, fermentation time, and germination days were selected for the RSM experiment based on a Box-Behnken center combination design (DTD), and the GABA level was treated as the response values. Table 1 shows the three factors and the three levels of the research design.

### 2.5. Determination of GABA Production by RP-HPLC

#### 2.5.1. Protein and Peptide Removal from Soybean Sprout Yogurt-like Product

GABA levels were determined according to the Wu and Shah’s method [2]. Reversed-Phase HPLC (RP-HPLC, Shimadzu model LC-2010A, Shimadzu Corp., Kyoto, Japan) was employed to detect the GABA concentration in the fermented soybean sprout yogurt-like product. First, to remove the protein of the soybean sprout milk, a 1 mL aliquot of fermented soymilk samples was diluted five times with sterile H_2_O, and 250 μL of zinc acetate and ferrous cyanide were added and mixed thoroughly. After standing for 1 h, samples were centrifuged by 5000 g at 25 °C to completely precipitate proteins and peptides. A GABA analysis was performed after samples were stored at 4 °C.

#### 2.5.2. Amino acid Derivatization 

The obtained supernatant containing GABA was derived. 200 μL of supernatant was mixed with 200 μL of acetonitrile, 200 μL of NaHCO_3_ (pH 9.8), 200 μL of H_2_O, and 100 μL 40 g L^−1^ of Dansyl chloride was added at 60 °C in the dark for 1 h. After derivation, 100 μL of 20 μL mL^−1^ acetic acids was added to stop the reaction. Subsequently, the sample was centrifuged at 12,000 g at 25 °C for 5 min. Moreover, the supernatant passed through a 0.22 μM filter with a membrane and was stored in a brown vial.

Subsequently, the GABA concentration of the derived sample was analyzed using RP-HPLC, as previously used [2]. The retention time for GABA is shown below at 20 min. Moreover, the standard curve of GABA present in Figure 1 was prepared with 0.01, 0.05, 0.07, 0.1, 0.25, 0.5, 0.7, and 1.75 g L^−1^ concentrations of GABA standard solution. It can be seen that the peak area was highly correlated with the GABA concentration, R^2^ = 0.9992, and the relationship between them satisfied the regression equation *y* = 4 × 10^6^ *x* + 70,120.

According to different experiments, the different integral areas obtained by the samples could be substituted into the formula GABA concentration. RP-HPLC was used to separate and quantify dansyl GABA and dansyl glutamic acid using a Kromasil 5-μm 100A C18 column (250 mm × 4.6 mm; Phenomenex, Torrance, CA, USA).

### 2.6. Determination of pH and Viable Cell Counts in Fermented Soybean Sprout Yogurt-like Product

This method was a combination of Chan and Wu’s research respectively [20,21]. The pH values of the fermented soybean sprout yogurt were measured using a pH meter (250 A Orion Portable pH Meter, US). To measure the viable cell number, 1 mL of the fermented soybean sprout yogurt-like product was dissolved in 9 mL of sterilized normal saline. Subsequently, 1 mL of the uniform solution was taken into Difeo TM lactobacilli MRS broth and incubated at 37 °C for 48 h. The average number of colonies in the Petri dish was multiplied and calculated by the dilution multiple. Generally, 30–300 CFU were chosen to count. The unit of colony numbers was CFU mL^−1^.

### 2.7. Protein Content of Fermented Soybean Sprout Yogurt-like Product

The fermented soybean sprout yogurt-like product was compared to those made from commercial soybean powder and milk powder. The protein content was determined using the MicroKjeldahl method. A sample of 0.3 g was accurately weighed and then placed in a Kjeldahl tube with a catalyst tablet and 10 mL of concentrated sulfuric acid. 

The weighed sample was nitrified in a nitrification furnace at 370 °C for 50 min until the solution turned light green. Next, 40 mL of distilled water was added to the nitrated sample for cooling and then put into the MicroKjeldahl nitrogen determinator for automatic titration. Finally, manual titration was conducted using a 250 mL conical flask with 40 mL of 4% boric acid solution and five drops of the indicator. Three parallel tests were performed for each group of samples. The sample nitrogen content was calculated using the following formula and then converted to crude protein content:(1)%N=((1.4×V)÷1000)⁄g )×100
*V* = volume (mL) of 0.1 N HCl used in the titration. The value of 1.4 is derived from the fact that 1.0 mL of 0.1 M NH_4_OH contains 1.4 mg nitrogen.

### 2.8. Texture Analysis

To compare the effects of different soybean germination days, fermentation time and carbon source additions on the texture of the yogurt-like products, a texture analysis was performed on samples with different preparation conditions. The method for textural characterization of the fermented bean sprout yogurt-like product was modified according to Giri’s method [22]. Before measurement, products with different preparation processes were stored at 4 °C for 12 h, restored to room temperature, and about 9 g of samples were weighed. The texture analysis was performed using a Texture Analyzer TAXT2i (Stable Micro Systems, Godalming, Surrey, UK) equipped with a 25 kg load cell and calibrated with a standard dead weight of 5 kg before use. A HDP/SR-TTC probe was unitized for determination. A Texture Expert version 1.20 (Stable Micro Systems) software application measured firmness, stickiness, work of shear, and work of adhesion. The same sample was weighed three times, and the mean value was obtained and recorded. 

The specific measurement parameters were test speed: 3.0 mm s^−1^, measured speed: 10 mm s^−1^, test distance: 23 mm, trigger force: g, and data acquisition rate: 200 PPS.

### 2.9. Sensory Analysis

This method was slightly modified from Meilgaard’s approach [23]. Briefly, 50 trained panelists were invited to evaluate the appearance, odor, acidity, thickness, fluidity, taste, and overall acceptance of the fermented yogurt-like product using 9-point scores (from 1 to 9). The ratings were presented on a 9-point hedonic scale ranging from 9 (“extremely like”) to 1 (“extremely dislike”).

### 2.10. Statistical Analysis

The data figure was created using the program, Microsoft Excel 2010, and IBM SPSS 25 Statistic was used to analyze the significant differences (*p* < 0.05 showed that the difference in the analysis results was significant, and *p* < 0.01 indicated that the difference in the analysis results was very significant). The response surface was designed, optimized, and analyzed using Design Export 10.0.7. 

## 3. Results and Discussion

### 3.1. The Effect of Various Conditions on GABA Production by Lb. brevis 145 in Soybean Sprout Yogurt-like Product

Legumes primarily metabolize GABA through a short pathway known as a GABA shunt, converting glutamate into succinic acid. This pathway synthesizes GABA from glutamate-by-glutamate decarboxylase (GAD, EC 4.1.1.15). GABA is converted to succinic semialdehyde (SSA) using GABA aminotransferase (GABA-T, EC 2.6.1.19). Then the last step of the shunt pathway is to convert SSA to succinic acid using succinic semialdehyde dehydrogenase (SSADH, EC 1.2.2.16) [24,25]. In the present study, after soybean seed germination, protein was transformed into glutamate and polyamine, which provided a sufficient precursor substance for GABA enrichment. During soybean germination, the content of soluble sugar decreased, and the content of dry matter decreased with the extension of germination time. However, the content of reduced sugar, soluble protein, free amino acid, and GABA increased. 

Figure 2A shows that a significant increase (*p* < 0.05) in GABA content was detected during soybean germination. GABA content in soybeans initially increased and then decreased with increasing germination time. When the germination time was one day, the GABA content was the most extensive (0.025 g L^−1^). Compared to raw soybeans, the GABA content increased continuously and significantly by 1.61-fold by the end of germination at one day. This outcome is consistent with the findings of Vann’s study [8], which found that soybean germination significantly increased the GABA content in soybean sprouts. Meanwhile, a previous study [4] reported that the GABA content in germinated soybeans peaked on day 5, which conflicted with the present study. As mentioned above, increased GAD activity could be responsible for higher GABA content. The most suitable explanation for this differential phenomenon is that the GAD activity during germination was also influenced by germination temperature and germination approaches [26,27].

Conversely, Figure 2B shows that GABA-producing bacteria show different preferences for sugars, affecting their growth and GABA production. Compared to lactose, mannose, galactose, amylopectin, and fructose, glucose were significantly different in terms of improving GABA levels (*p* < 0.05). This result is consistent with the findings of a previous report by Xiao and Shah [19], which suggested that after fermentation for 24 h, glucose was the main carbon source consumed by *Lb. brevis* 145.

Furthermore, with increasing fermentation time, the GABA content in the fermentation broth increased initially and then decreased (Figure 2C). The content increased sharply in the first 48 h. After 48 h of continuous culture, the GABA content in the fermentation broth decreased significantly. At 48 h of fermentation, the content of GABA reached its maximum, which was 1.867 g L^−1^. A possible reason for this was the consumption of MSG and nutrients in the fermentation broth with the extension of fermentation time, and the subsequent cell senescence with decreased GABA content. 

An investigation of the effect of different glucose concentrations on GABA production in the soybean sprout yogurt-like product was also performed. Glucose, as the main carbon source of microorganisms, has the energy required for the life activities of bacteria, and constitutes the material basis of bacterial cells and their metabolites [28]. Figure 2D shows that with increasing glucose addition, the GABA content in the fermented bean sprout yogurt-like product also increased initially and then decreased. Briefly, when the glucose addition was 10 g L^−1^, the maximum GABA content was 2.21 g L^−1^. However, when the glucose addition continued to increase, the GABA content in the bean sprout yogurt-like product demonstrated an obvious decreasing trend. It could be that when the sugar content in the fermentation medium was too high, the cell metabolic activity produced organic acids, resulting in decreased pH and cell aging. Moreover, when the sugar content in the fermentation medium was in a low range, the bacteria were less affected by changes in sugar metabolites [28]. 

### 3.2. Optimizing the Fermentation Process to Produce GABA by Lb. brevis 145 in Soybean Sprout Yogurt-like Product

#### 3.2.1. RSM Results

The carbon concentration, fermentation times, and germination days were chosen as the key variables and the focal points for the response surface analysis in order to simulate the fermentation process based on the single-variable optimization (Table 2). Based on the Box-Behnken experimental design results, a quadratic multiple regression fitting was conducted, and a multiple quadratic response surface regression model was established. The obtained quadratic regression equation was as follows:*Y*= 2.35 + 5.9375 × 10^−3^ × *A* + 4.1875 × 10^−3^ × *B* + 5.125 × 10^−3^ × *C*−1.75 × 10^−3^ × *AB* + 3.75 × 10^−4^ × *AC* + 3.75 × 10^−4^ × *BC*−0.060135 × *A*^2^−0.046135 × *B*^2^−0.04801 × *C*^2^(2)
where *Y* represents the GABA concentration, *A* represents the germination days, *B* represents the fermentation time, and *C* represents the glucose concentration.

Table 3 illustrates the results of ANOVA. The regression model *F* test presented high significance (*p* < 0.01), and the R-squared was 95.75%, indicating that the model could explain the change in the 95.75% response value. The lack of fit was 0.676 (more than 0.05), which was non-significant. The model had a high degree of fit with the data and a small experimental error. This model and equation could be employed to analyze and predict the amount of GABA extraction.

#### 3.2.2. RSM Analysis of the Best-Fermented Parameters

Following the linear regression equation fitted by the RSM, the response surface graph and contour of the model were drawn. The response surface contour map directly reflected the influence of various factors on the response value to find out the best process parameters and the interaction between various parameters. The center point of the smallest ellipse in the contour was the highest point of response surface; the contour map shape reflected the intensity and significance of interaction between the two factors. The contour lines in Figure 3 were oval, corroborating the finding the interaction between fermentation time and the addition of sugar concentration to germination time was significant. 

Figure 3 presents the three-dimensional spatial surface diagram of the interaction of two factor independent variables on GABA concentration created by Design Expert 10.0.7 software. The 3D response surface diagrams show that germination days, glucose concentration, and fermentation time had a good interaction, and that their effects were all statistically significant. By analyzing the linear regression equation, it was found that there was a maximum point in the experiment, which was also the maximum point in this study. Technological conditions producing this maximum point could be found through response surface analysis. Thus, the optimal technological conditions for the enrichment of GABA from fermented soybean sprout yogurt-like product were: soybean germination for 0.798 days, fermentation time for 45.490 h, and glucose concentration of 9.691 g L^−1^. Under these conditions, the predicted value of the GABA mass concentration was 2.287 g L^−1^. In order to verify the reliability of the regression equation, under the optimized conditions, soybeans germinated for 24 h, fermented for 48 h, and 10 g L^−1^ of glucose concentration was adopted; the GABA level obtained from the verification test was 2.302 g L^−1^, and the relative deviation was 0.67% compared to the theoretical prediction value. Therefore, the optimal process conditions of the fermentation system obtained by the response surface optimization method were reliable. Furthermore, the fermented soybean sprout yogurt-like product obtained in the validation test had a uniform solidification state, a strong fermentation flavor, a pure flavor, and no peculiar smell.

### 3.3. Finished Product Quality Analysis

#### 3.3.1. GABA Concentration in GABA-Rich Yogurt

Figure 4 shows that the soybeans were treated with *Lb. brevis* 145 after 48 h of fermentation after one day of germination. The GABA content reached a maximum of 2.302 g L^−1^, which was 1.56 and 3.5 times the GABA content in yogurt-like product fermented with soybean powder and milk powder respectively, implying that soybean germination and fermentation of lactic acid bacteria could significantly increase the GABA content in yogurt, thus producing a functional yogurt-like product rich in GABA.

#### 3.3.2. pH and Cell Viability in Fermented Soybean Sprout Yogurt-like Product

According to the Chinese national standard, GB 4789.35, for lactic acid bacteria content in viable products, the lactic acid bacteria content must be higher than 1 × 10^6^ CFU mL^−1^. Figure 5 illustrates that the number of bacteria after 72 h of fermentation, still up to 8 × 10^6^ CFU mL^−1^, already met the standard requirement for live bacteria plant yogurt. 

Moreover, as the fermentation time increases, acidity elevates due to the production of organic acids in the medium, resulting in a decrease in pH. After fermentation, the pH of the soybean sprout yogurt-like product also showed a downward trend, as illustrated in Figure 5. Finally, the pH of the fermented bean sprout yogurt-like product was maintained at about 4.4, which meets the Chinese national standard requirement (GB 5009.237) for the pH of fermented yogurt products (pH ≤ 4.5). 

#### 3.3.3. Texture Characteristic and Protein Content in GABA-Rich Yogurt-like Product

Table 4 illustrates the texture characteristics and protein content of fermented yogurt-like product from soybean sprouts, fermented yogurt-like product from soy flour, and fermented yogurt from milk. The texture samples were obtained from samples with intact gel structures after 48 h of fermentation; their work of shear, stickiness, work of adhesion, and firmness was evaluated. 

Firmness, or the force required to achieve a certain deformation, is a regularly examined criterion when defining the texture of set-type cultured dairy products. It is the peak force height on the first compression cycle [29,30]. The firmness of fermented soybean sprout yogurt-like product was significantly (*p* < 0.05) higher than the other two samples. The increased firmness could be due to the high water binding capacity [31,32]. 

The quantity of energy required to perform the shear operation is known as the work of shear. It therefore evaluates the sample resistance throughout the penetration of the probe. In the current investigation, the work of shear of the fermented soybean sprout yogurt-like product and the fermented soybean flour yogurt-like product was significantly (*p* < 0.05) higher than that of the fermented milk yogurt. However, no significant (*p* > 0.05) difference was observed between the fermented soybean sprout yogurt-like product and the fermented soybean flour yogurt-like product.

Stickiness is an essential sensory quality of semisolid food ingredients, defined as a sensation sensed by the tongue and palate [33,34]. Negative stickiness values represent stickiness, while positive values represent the product’s hardness. In the present investigation, no significant (*p* > 0.05) difference in stickiness was detected between the fermented soybean sprout yogurt-like product and the fermented soybean flour yogurt-like product, both of which were higher levels of stickiness than the fermented milk yogurt.

To characterize the work of adhesion, the area under the negative peak in penetration was measured. It can also be defined as the work required to overcome the attraction force between the product surface and the probe surface [22]. During the current investigation, as Table 4 illustrates, there was no significant difference in the work of adhesion between the fermented soybean sprout yogurt-like product and the fermented soy flour yogurt-like product (*p* > 0.05), both of which were marginally lower compared to the fermented milk yogurt.

Furthermore, according to the Chinese national standard requirement (GB 5009.5), the protein content in soybean products is not allowed to be less than 2.5%. Table 4 shows that the protein content of the three products reached the national standard. These results were consistent with the result of Niamah’s study [35]. Notably, the protein level of the fermented soybean sprout yogurt-like product exceeded the national standard by 1.7 times.

#### 3.3.4. Sensory Evaluation of Yogurt-like Product Rich in GABA

Table 5 shows the scores for the sensory characteristics of the fermented samples. GABA-rich fermented sprout yogurt-like product had a milky and full-bodied aroma. There were no significant differences (*p* > 0.05) in appearance, acidity, fluidity, thickness, or overall acceptance between the bean sprout yogurt-like product and the commercially available yogurt, validating the finding that fermented bean sprout yogurt-like product has prospective market acceptance and consumer acceptance. However, there was a significant difference (*p* < 0.05) in odor and taste between the bean sprout yogurt-like product and the commercially available yogurt. Future process optimization will focus on improving these two indicators. 

## 4. Conclusions

This study investigated the effect of lactic acid bacteria fermentation of germinated soybeans on the GABA content of yogurt. In soybeans, GABA content increases significantly during the germination and reaches its peak after one day of germination. The highest level of GABA production (2.302 g L^−1^) of the fermented soybean sprout yogurt-like product was obtained when *Lb. brevis* 145 was fermented with glucose 10 g L^−1^ as the sole carbon source for 48 h. The use of germinated soybeans had a significantly positive effect on GABA enrichment. Simultaneously, the fermented soybean sprout yogurt-like product with high GABA content met the requirements of Chinese national standards for yogurt in terms of acidity, protein content, and the number of live bacteria, and it had a better texture than the commercially available yogurt. This provides a prerequisite for producing innovative GABA-enriched yogurt.

## Figures and Tables

**Figure 1 foods-12-00977-f001:**
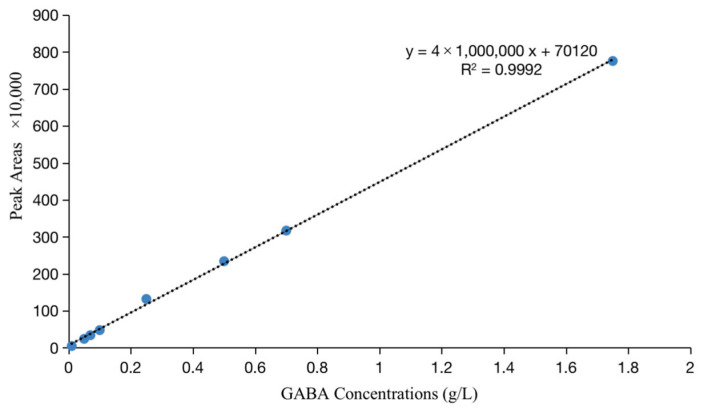
The standard curve of GABA in different solutions and peak areas.

**Figure 2 foods-12-00977-f002:**
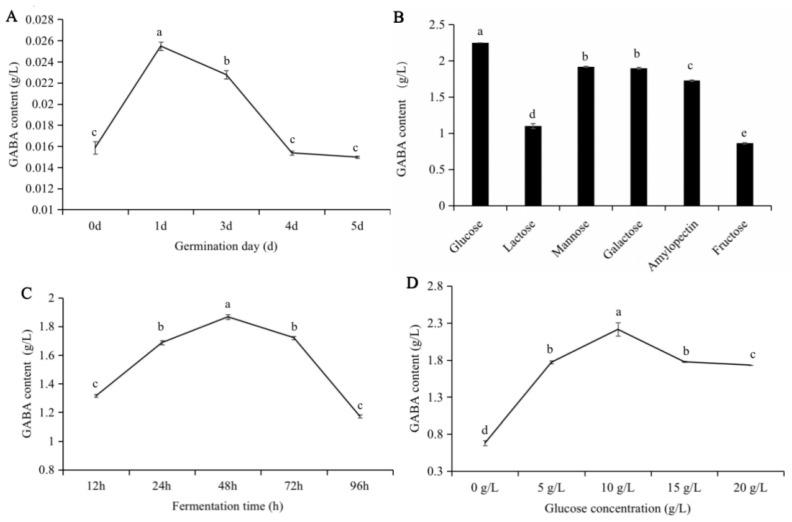
Effect of different conditions on the production of GABA by *Lb. brevis* 145 in soybean sprout yogurt. (**A**) represents germination day; (**B**) represents carbon source types; (**C**) represents fermentation time; (**D**) represents glucose concentration. Means and standard deviations of triplicate experimental data were represented by the values and error bars. a, b, c, d, and e in the figure represent different product were significant difference in various conditions.

**Figure 3 foods-12-00977-f003:**
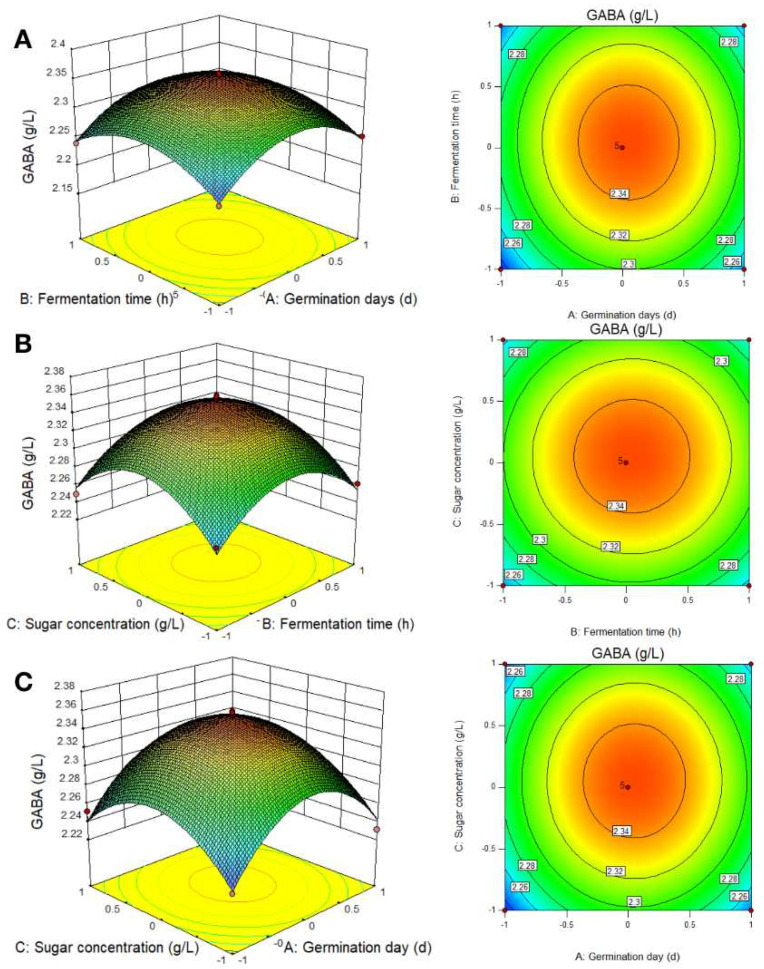
Three-dimensional surface plots show the effects of different variables on GABA production. (**A**) Effect of germination days and fermentation time; (**B**) Effect of fermentation time and sugar concentration; (**C**) Effect of sugar concentration and germination day.

**Figure 4 foods-12-00977-f004:**
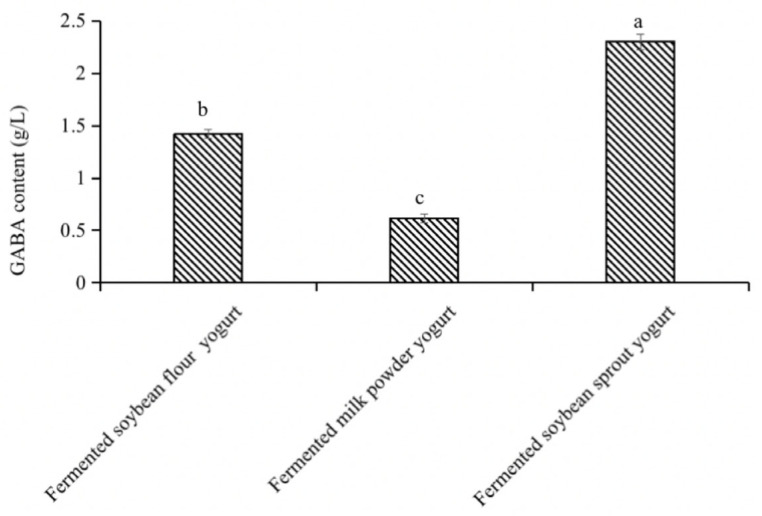
GABA content in different types of yogurts. a, b, and c in the figure represent GABA level of three products were significantly different.

**Figure 5 foods-12-00977-f005:**
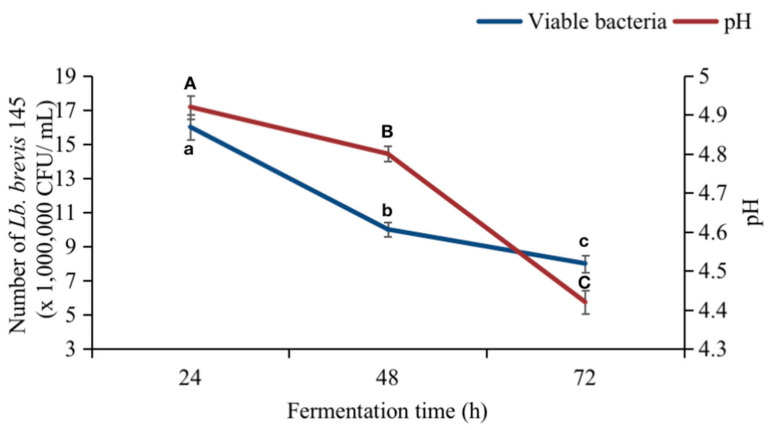
Changes in viable bacteria and pH in fermented bean sprout yogurt-like product with fermentation time. a, b, c, and A, B, C in the figure represent significant differences in yogurt-like products with increasing fermentation time, respectively.

**Table 1 foods-12-00977-t001:** Factors and levels of response surface analysis.

Encoding	A: Germination Day (d)	B: Fermentation Time (h)	C: Sugar Concentration (g L^−1^)
−1	0	24	5
0	1	48	10
1	3	72	15

**Table 2 foods-12-00977-t002:** Experimental design of Box-Behnken and corresponding results.

No.	A	B	C	GABA Conc (g L^−1^)
1	−1	−1	0	2.228
2	1	−1	0	2.252
3	−1	1	0	2.240
4	1	1	0	2.256
5	−1	0	−1	2.229
6	1	0	−1	2.232
7	−1	0	1	2.252
8	1	0	1	2.256
9	0	−1	−1	2.254
10	0	1	−1	2.262
11	0	−1	1	2.250
12	0	1	1	2.259
13	0	0	0	2.319
14	0	0	0	2.356
15	0	0	0	2.360
16	0	0	0	2.358
17	0	0	0	2.358

**Table 3 foods-12-00977-t003:** GABA production regression analysis using the Box-Behnken DTD experimental design.

Source of Mean Square	Sum of Squares	Degree of Freedom	Mean Square	*F*	Pr > *F*	Significance
Model	0.038444824	9	0.004271647	17.50715543	0.000525758	Significant
A	0.000282031	1	0.000282031	1.15589251	0.317973848	ns
B	0.000140281	1	0.000140281	0.574936452	0.47305463	ns
C	0.000210125	1	0.000210125	0.861187949	0.384288761	ns
AB	1.225 × 10^−5^	1	1.225 × 10^−5^	0.050206079	0.829104465	ns
AC	5.625 × 10^−7^	1	5.625 × 10^−7^	0.002305381	0.963045971	ns
BC	5.625 × 10^−7^	1	5.625 × 10^−7^	0.002305381	0.963045971	ns
A^2^	0.015226182	1	0.015226182	62.40382843	9.87946E−05	**
B^2^	0.008961845	1	0.008961845	36.72972303	0.000510618	*
C^2^	0.009705095	1	0.009705095	39.77590003	0.000401524	*
Residual	0.001707961	7	0.000243994			
Lack of Fit	0.000496813	3	0.000165604	0.546932882	0.67624669	ns
Pure Error	0.001211148	4	0.000302787			
Cor Total	0.040152785	16				

R-squared = 95.75%. ns represent there was no significant difference in the statistical result; * represent there was significant difference in the statistical result (*p* < 0.05); ** represent there was significant difference in the statistical result (*p* < 0.01)

**Table 4 foods-12-00977-t004:** Texture characteristics and protein content of fermented bean sprout yogurt-like product. a, b, and c in the table represent three products were significant difference in various textural analysis.

Item	Texture Analysis				Protein (%)
	Firmness	Work of Shear	Stickiness	Work of Adhesion	
Fermented bean sprout yogurt-like product	8.81 ± 0.31 ^a^	4.4 ± 0.21 ^a^	−7.85 ± 0.07 ^b^	−11.55 ± 0.13 ^b^	4.41 ± 0.05 ^a^
Fermented soybean flour yogurt-like product	4.55 ± 0.29 ^b^	4.22 ± 0.19 ^a^	−7.97 ± 0.11 ^b^	−11.26 ± 0.31 ^b^	2.93 ± 0.31 ^c^
Fermented milk yogurt	5.78 ± 0.42 ^c^	1.93 ± 0.07 ^b^	−1.53 ± 0.27 ^a^	−1.05 ± 0.31 ^a^	3.22 ± 0.17 ^b^

**Table 5 foods-12-00977-t005:** Sensory characteristics of fermented yogurts (means ± SD for *n* = 3). a and b in the table represent two product were significant difference in various sensory evaluation indice.

Item	Soybean Sprout Yogurt-Like Product	Commercially Available Yogurt
Appearance	7.75 ± 0.56 ^a^	7.88 ± 0.48 ^a^
Odor	5.63 ± 0.41 ^b^	7.75 ± 0.29 ^a^
Acidity	7.25 ± 0.25 ^a^	7.25 ± 0.29 ^a^
Thickness	7.63 ± 0.22 ^a^	7.75 ± 0.41 ^a^
Fluidness	7.15 ± 0.25 ^a^	7.25 ± 0.41 ^a^
Taste	6.75 ± 0.35 ^b^	7.13 ± 0.48 ^a^
Overall acceptance	7.75 ± 0.41 ^a^	7.75 ± 0.29 ^a^

## Data Availability

Not applicable.

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
