# Peer review of "Optimizing Levilactobacillus brevis NPS-QW 145 Fermentation for Gamma-Aminobutyric Acid (GABA) Production in Soybean Sprout Yogurt-like Product"

_foods, 2023, doi:10.3390/foods12050977_

Round 1

Reviewer 1 Report

Dear Editors and authors,

1-The naming of the species and genera of lactic acid bacteria changed three years ago. Why do you use the old naming for this bacterium? Authors must, when writing any manuscript, refer to recent references , see tittle and line 37. The new name of Lactobacillus Brevis is Levilactobacillus brevis.

2-How did the starter culture used in the production of soybean yoghurt become active? You should mention the method I suggest you read (Niamah, A. K., Sahi, A. A., & Al-Sharifi, A. S. (2017). Effect of feeding soy milk fermented by probiotic bacteria on some blood criteria and weight of experimental animals. Probiotics and Antimicrobial Proteins9(3), 284-291).

3-What is the number of bacteria in the 3% inoculum used in the production of soybean yogurt?

4-The Optimum temperature of this bacteria is 30C, why used 37 C in soy fermentation ? see line 79 and 138.

5-The author used several conditions in order to choose the optimal conditions for production, while the temperature remained at 37 °C, which is not the Optimum degree for the growth of bacteria. Why did you not use different temperatures for fermentation??? 

6-Protein contents of fermented soybean sprout yogurt needs to add new references , I suggest  Bremner, J. M. (1960). Determination of nitrogen in soil by the Kjeldahl method. The Journal of Agricultural Science55(1), 11-33.

7-The researcher used the Kjeldahl method to estimate the protein, but he took 0.3 g of the sample, and this shows that the method is MicroKjeldahl method.

8-Figure 2b The name of the x-axis should be written, Carbon sources.

9-Figure 5 needs to make adjustments. The y-axis title should write Number of L. brevis  (CFU/ ml), What do the uppercase and lowercase letters mean.

10-Conclusions contain many results, you must rewrite the conclusions again and delete all results from it.

Author Response

Dear Editor:
We gratefully appreciate the editors and all reviewers for their time and making positive and constructive comments. These comments are all valuable and helpful for revising and improving our manuscript entitled “Optimizing Levilactobacillus brevis NPS-QW 145 fermentation for Gamma-aminobutyric acid (GABA) production in soybean sprout yogurt” (ID: 2145380), as well as the important guiding significance to our researches.
We have studied comments carefully and have made correction which we hope meet with approval. Revised portion are marked in red in the revised manuscript. The summary of corrections and the responses to the reviewer’s comments are listed in the Revision Report.

Thank you and best wishes
Yours sincerely,
ZHANG Yue
Corresponding author:
Name: Xiao Chaogeng & Nagendra P. Shah 
E-mail: xiaochaogeng@163.com; npshah@hku.hk

Revision report
First of all, I would like to express our sincere gratitude to the reviewers for their comments. These comments are all valuable and helpful for revising and improving our manuscript, as well as the important guiding significance to our researches. We have studied comments carefully and have made correction which we hope meet with approval. Revised portion are marked in red in the revised manuscript. The summary of corrections and the responses to the reviewer’s comments are listed below.
Summary of the revision:
Section 1(Name): Updated the name of lactic acid bacteria
Section 2 (Modification): Corresponding parts of the article were revised according to the reviewers' suggestions
Section 3 (Conclusion): The overall conclusion section was rewritten to incorporate the comments received.
Section 4 (Checking rate): The overall reduction of repetition rate was carried out in the article.

In addition, the manuscript has been carefully revised considering the language and grammar problems, including the sentences that are convoluted and hard to follow.

Reviewer 2 Report

The manuscript needs subject to major revisions.  I have included all detailed comments in the attached pdf file.

Author Response

Revision report

First, I would like to express our sincere gratitude to the reviewers for their comments. These comments are all valuable and helpful for revising and improving our manuscript, as well as the important guiding significance to our research. We have studied comments carefully and have made correction which we hope meet with approval. Revised portion are marked in red in the revised manuscript. The summary of corrections and the responses to the reviewer’s comments are listed below.

Summary of the revision:

Section 1(Name): Updated the name of lactic acid bacteria

Section 2 (Modification): Corresponding parts of the article were revised according to the reviewers' suggestions

Section 3 (Conclusion): The overall conclusion section was rewritten to incorporate the comments received.

Section 4 (Checking rate): The overall reduction of repetition rate was carried out in the article.

In addition, the manuscript has been carefully revised considering the language and grammar problems, including the sentences that are convoluted and hard to follow.

Reviewer 3 Report

This is an interesting study that has investigated the effect of lactic acid bacteria fermentation of germinated soybeans on the GABA content of yogurt and has proposed optimum conditions for the production of a  new GABA-enriched yogurt.

Some remarks for improvement:

Title: Please use italics in all names of microorganisms and also please follow the correct nomenclature of microorganims eg. the name of the species (brevis) is always starting with small letter not capital letter.

Moreover, Lactobacillus brevis, is no longer called with that name because it has been reclassified to another genus: Levilactobacillus brevis so please make the necessary corrections

Line 37: the name of Lactobacillus paracasei has changed to Lacticaseibacillus paracasei

Line 77: it would be useful to know the number of viable microorganisms in the inoculum

Line 78: please state the different concentrations

Line 237: In Figure 2, it is not clear for each graph presented what were the values for all the other variables tested, eg in graph B that shows the GABA production per carbon source, in what conditions was this graph produced? (during a fermentation?, with the presence of LAB? ), the same questions apply to all graphs.

Author Response

Dear Editor:

We gratefully appreciate the editors and all reviewers for their time and making positive and constructive comments. These comments are all valuable and helpful for revising and improving our manuscript entitled “Optimizing Levilactobacillus brevis NPS-QW 145 fermentation for Gamma-aminobutyric acid (GABA) production in soybean sprout yogurt” (ID: 2145380), as well as the important guiding significance to our researches.

We have studied comments carefully and have made correction which we hope meet with approval. Revised portion are marked in red in the revised manuscript. The summary of corrections and the responses to the reviewer’s comments are listed in the Revision Report.

Thank you and best wishes

Yours sincerely,

ZHANG Yue

Name: Xiao Chaogeng & Nagendra P. Shah 

E-mail: xiaochaogeng@163.com; npshah@hku.hk

Revision report

First of all, I would like to express our sincere gratitude to the reviewers for their comments. These comments are all valuable and helpful for revising and improving our manuscript, as well as the important guiding significance to our researches. We have studied comments carefully and have made correction which we hope meet with approval. Revised portion are marked in red in the revised manuscript. The summary of corrections and the responses to the reviewer’s comments are listed below.

Summary of the revision:

Section 1 (Name): Updated the name of lactic acid bacteria

Section 2 (Modification): Corresponding parts of the article were revised according to the reviewers' suggestions

Section 3 (Conclusion): The overall conclusion section was rewritten to incorporate the comments received.

Section 4 (Checking rate): The overall reduction of repetition rate was carried out in the article.

In addition, the manuscript has been carefully revised considering the language and grammar problems, including the sentences that are convoluted and hard to follow.

Responses to reviewers

Reviewer #3:

Point 1: Please use italics in all names of microorganisms and also please follow the correct nomenclature of microorganims eg. the name of the species (brevis) is always starting with small letter not capital letter. Moreover, Lactobacillus brevis, is no longer called with that name because it has been reclassified to another genus: Levilactobacillus brevis so please make the necessary corrections.

Response 1: Thank you for your guidance, I made corresponding modifications in the article. I changed Lactobacillus brevis to Levilactobacillus brevis.

Point 2: Line 37: the name of Lactobacillus paracasei has changed to Lacticaseibacillus paracasei

Response 2: Thank you for your guidance, I made corresponding modifications in the article.

“such as Levilactobacillus brevis, Lacticaseibacillus paracasei, and Enterococcus raffinosu

Point 3: Line 77: it would be useful to know the number of viable microorganisms in the inoculum.

Response 3: Thank you for your valuable advice. It is quite difficult to measure the number of live bacteria in it without fermentation. And our research was based on the previous fermentation experiments, which were inoculated with 3% Lb. brevis 145. In addition, I also measured the number of viable bacteria after fermentation, and the number of viable bacteria can reach 1.6×107 CFU/mL at 24 hours of fermentation.

This is the reference:

Xiao, T., & Shah, N. P. (2021). Lactic acid produced by Streptococcus thermophilus activated glutamate decarboxylase (GadA) in Lactobacillus brevis NPS-QW 145 to improve γ-amino butyric acid production during soymilk fermentation. LWT, 137, 110474.

Point 4: Line 78: please state the different concentrations

Response 4: Thank you for your suggestion. The different concentrations I marked in the original manuscript.

“5 g L-1 MSG, and six different monosaccharides (glucose, lactose, mannose, galactose, amylopectin, and fructose) at different concentrations (0, 5, 10, 15, and 20 g L-1) and fermented in the 37 ℃ incubator to observe the coagulation state and compared the GABA concentration in the yogurt. ”

Point 5: Line 237: In Figure 2, it is not clear for each graph presented what were the values for all the other variables tested, eg in graph B that shows the GABA production per carbon source, in what conditions was this graph produced? (during a fermentation?, with the presence of LAB? ), the same questions apply to all graphs.

Response 5: Thank you for your kind advice. Figure A shows the study of the effect of germination time on the GABA content in fermented bean sprout milk, ensuring that other conditions are constant, i.e. the carbon source used was glucose at a concentration of 10 g/L and fermented for 48 h. On the same principle, Figure B shows the study of the bioavailability of Lb. brevis 145 to different carbon sources, and Figure C shows the study of the effect of different fermentation times on the GABA content in fermented bean sprout milk. Figure D is a study of the effect of different glucose concentrations on the GABA content in fermented bean sprout milk.

Round 2

Reviewer 1 Report

Dear Editors 

The authors made all necessary changes to improve the manuscript, and now I recommend it for publication in its current form.

Author Response

No.: foods-2145380
Title: Optimizing Levilactobacillus brevis NPS-QW 145 Fermentation for Gamma-Aminobutyric Acid (GABA) Production in Soybean Sprout Yogurt-like Product
Dear Editor:

We gratefully appreciate the editors and all reviewers for their time and making positive and constructive comments. These comments are all valuable and helpful for revising and improving our manuscript entitled “Optimizing Levilactobacillus brevis NPS-QW 145 Fermentation for Gamma-Aminobutyric Acid (GABA) Production in Soybean Sprout Yogurt-like Product” (ID: foods-2145380), as well as the important guiding significance to our researches.

We have studied comments carefully and have made correction which we hope meet with approval. Revised portion are marked in red in the revised manuscript. The summary of corrections and the responses to the reviewer’s comments are listed in the Revision Report.

Thank you and best wishes

Yours sincerely,

ZHANG Yue

Name: Xiao Chaogeng&Nagendra P. Shah 

E-mail: xiaochaogeng@163.com; npshah@hku.hk

Reviewer 2 Report

The authors complied with the reviewer's comments. Nevertheless, the following needs to be supplemented:

1. Correction of references in accordance with the requirements of the journal and some spelling mistakes: genus and species names of bacteria are written in italics.

2. Add an explanation of the statistical analysis under each figure and table as I requested in review (explain what the letters abc etc mean).

Author Response

No.: foods-2145380
Title: Optimizing Levilactobacillus brevis NPS-QW 145 Fermentation for Gamma-Aminobutyric Acid (GABA) Production in Soybean Sprout Yogurt-like Product
Dear Editor:

We gratefully appreciate the editors and all reviewers for their time and making positive and constructive comments. These comments are all valuable and helpful for revising and improving our manuscript entitled “Optimizing Levilactobacillus brevis NPS-QW 145 Fermentation for Gamma-Aminobutyric Acid (GABA) Production in Soybean Sprout Yogurt-like Product” (ID: foods-2145380), as well as the important guiding significance to our researches.

We have studied comments carefully and have made correction which we hope meet with approval. Revised portion are marked in red in the revised manuscript. The summary of corrections and the responses to the reviewer’s comments are listed in the Revision Report.

Thank you and best wishes

Yours sincerely,

ZHANG Yue

Name: Xiao Chaogeng&Nagendra P. Shah 

E-mail: xiaochaogeng@163.com; npshah@hku.hk

Revision Report

Editor's comments:

The authors complied with the reviewer's comments. Nevertheless, the following needs to be supplemented:

Point 1. Correction of references in accordance with the requirements of the journal and some spelling mistakes: genus and species names of bacteria are written in italics.

Response 1: Thank you for your advice. I corrected the reference style, such as italics the name of bacteria, name of journal.

References

[1] Wu, Q., & Shah, N. P. (2015). Gas release-based prescreening combined with reversed-phase HPLC quantitation for efficient selection of high-γ-aminobutyric acid (GABA)-producing lactic acid bacteria. Journal of dairy science, 98(2), 790-797.

[2] Wu, Q., & Shah, N. P. (2018). Restoration of GABA production machinery in Lactobacillus brevis by accessible carbohydrates, anaerobiosis and early acidification. Food microbiology, 69, 151-158.

[3] Binh, T. T. T., Ju, W. T., Jung, W. J., & Park, R. D. (2014). Optimization of γ-amino butyric acid production in a newly isolated Lactobacillus brevis. Biotechnology letters, 36(1), 93-98.

[4] Luo, X., Wang, Y., Li, Q., Wang, D., Xing, C., Zhang, L., Xu, T., Fang, F., & Wang, F. (2018). Accumulating mechanism of γ‐aminobutyric acid in soybean (Glycine max L.) during germination. International Journal of Food Science & Technology, 53(1), 106-111.

[5] Wang, Y., Liu, C., Ma, T., & Zhao, J. (2019). Physicochemical and functional properties of γ-aminobutyric acid-treated soy proteins. Food chemistry, 295, 267-273.

[6] Park, K. B., & Oh, S. H. (2007). Production of yogurt with enhanced levels of gamma-aminobutyric acid and valuable nutrients using lactic acid bacteria and germinated soybean extract. Bioresource Technology, 98(8), 1675-1679.

[7] Shan, Y., Man, C. X., Han, X., Li, L., Guo, Y., Deng, Y., Li, T., Zhang, L. W., & Jiang, Y. J. (2015). Evaluation of improved γ-aminobutyric acid production in yogurt using Lactobacillus plantarum NDC75017. Journal of dairy science, 98(4), 2138-2149.

[8] Vann, K., Techaparin, A., & Apiraksakorn, J. (2020). Beans germination as a potential tool for GABA-enriched tofu production. Journal of Food Science and Technology, 57(11), 3947-3954.

[9] Tung, Y. T., Lee, B. H., Liu, C. F., & Pan, T. M. (2011). Optimization of culture condition for ACEI and GABA production by lactic acid bacteria. Journal of food science, 76(9), M585-M591.

[10] Aoki, H., Furuya, Y., Endo, Y., & Fujimoto, K. (2003). Effect of γ-aminobutyric acid-enriched tempeh-like fermented soybean (GABA-tempeh) on the blood pressure of spontaneously hypertensive rats. Bioscience, Biotechnology, and Biochemistry, 67(8), 1806-1808.

[11] Geisler, C. E., Ghimire, S., Bruggink, S. M., Miller, K. E., Weninger, S. N., Kronenfeld, J. M., Yoshino, J., Klein, S., Duca, F. A., & Renquist, B. J. (2021). A critical role of hepatic GABA in the metabolic dysfunction and hyperphagia of obesity. Cell reports, 35(13), 109301.

[12] Abd El-Fattah, A., Sakr, S., El-Dieb, S., & Elkashef, H. (2018). Developing functional yogurt rich in bioactive peptides and gamma-aminobutyric acid related to cardiovascular health. LWT, 98, 390-397.

[13] Sharma, S., Saxena, D. C., & Riar, C. S. (2018). Changes in the GABA and polyphenols contents of foxtail millet on germination and their relationship with in vitro antioxidant activity. Food Chemistry, 245, 863-870.

[14] Zhang, L., Yue, Y., Wang, X., Dai, W., Piao, C., & Yu, H. (2022). Optimization of fermentation for γ-aminobutyric acid (GABA) production by yeast Kluyveromyces marxianus C21 in okara (soybean residue). Bioprocess and Biosystems Engineering, 1-13.

[15] Ohmori, T., Tahara, M., & Ohshima, T. (2018). Mechanism of gamma-aminobutyric acid (GABA) production by a lactic acid bacterium in yogurt-sake. Process biochemistry, 74, 21-27.

[16] Guo, Y., Yang, R., Chen, H., Song, Y., & Gu, Z. (2012). Accumulation of γ-aminobutyric acid in germinated soybean (Glycine max L.) in relation to glutamate decarboxylase and diamine oxidase activity induced by additives under hypoxia. European Food Research and Technology, 234(4), 679-687.

[17] Ma, Y., Wang, P., Gu, Z., Sun, M., & Yang, R. (2022). Effects of germination on physio-biochemical metabolism and phenolic acids of soybean seeds. Journal of Food Composition and Analysis, 112, 104717.

[18] Sęczyk, Ł., Świeca, M., & Gawlik-Dziki, U. (2017). Soymilk enriched with green coffee phenolics–Antioxidant and nutritional properties in the light of phenolics-food matrix interactions. Food Chemistry, 223, 1-7.

[19] Xiao, T., & Shah, N. P. (2021). Lactic acid produced by Streptococcus thermophilus activated glutamate decarboxylase (GadA) in Lactobacillus brevis NPS-QW 145 to improve γ-amino butyric acid production during soymilk fermentation. LWT, 137, 110474.

[20] Chan, C. L., Gan, R. Y., Shah, N. P., & Corke, H. (2018). Enhancing antioxidant capacity of Lactobacillus acidophilus-fermented milk fortified with pomegranate peel extracts. Food bioscience, 26, 185-192.

[21] Wu, Q., Law, Y. S., & Shah, N. P. (2015). Dairy Streptococcus thermophilus improves cell viability of Lactobacillus brevis NPS-QW-145 and its γ-aminobutyric acid biosynthesis ability in milk. Scientific reports, 5(1), 1-12.

[22] Giri, A., Kanawjia, S. K., & Khetra, Y. (2014). Textural and melting properties of processed cheese spread as affected by incorporation of different inulin levels. Food and bioprocess technology, 7(5), 1533-1540.

[23] Meilgaard, M. C., Carr, B. T., & Civille, G. V. (1999). Sensory evaluation techniques. CRC press.

[24] Rizzello, C. G., Cassone, A., Di Cagno, R., & Gobbetti, M. (2008). Synthesis of angiotensin I-converting enzyme (ACE)-inhibitory peptides and γ-aminobutyric acid (GABA) during sourdough fermentation by selected lactic acid bacteria. Journal of agricultural and food chemistry, 56(16), 6936-6943.

[25] Yang, R., Feng, L., Wang, S., Yu, N., & Gu, Z. (2016). Accumulation of γ‐aminobutyric acid in soybean by hypoxia germination and freeze–thawing incubation. Journal of the Science of Food and Agriculture, 96(6), 2090-2096.

[26] Hwang, C. E., Haque, M., Lee, J. H., Song, Y. H., Lee, H. Y., Kim, S. C., & Cho, K. M. (2018). Bioconversion of γ-aminobutyric acid and isoflavone contents during the fermentation of high-protein soy powder yogurt with Lactobacillus brevis. Applied Biological Chemistry, 61(4), 409-421.

[27] Xu, J. G., & Hu, Q. P. (2014). Changes in γ-aminobutyric acid content and related enzyme activities in Jindou 25 soybean (Glycine max L.) seeds during germination. LWT, 55(1), 341-346.

[28] Chew, S. Y., & Than, L. T. L. (2021). Glucose Metabolism and Use of Alternative Carbon Sources in Medically-Important Fungi. Encyclopedia of Mycology, 2021, 220-229

[29] Meena, P. K., Gupta, V. K., Meena, G. S., Raju, P. N., & Parmar, P. T. (2015). Application of ultrafiltration technique for the quality improvement of dahi. Journal of food science and technology, 52(12), 7974-7983.

[30] Buriti, F. C., Castro, I. A., & Saad, S. M. (2010). Effects of refrigeration, freezing and replacement of milk fat by inulin and whey protein concentrate on texture profile and sensory acceptance of synbiotic guava mousses. Food Chemistry, 123(4), 1190-1197.

[31] Chen, L., Alcazar, J., Yang, T., Lu, Z., & Lu, Y. (2018). Optimized cultural conditions of functional yogurt for γ-aminobutyric acid augmentation using response surface methodology. Journal of dairy science, 101(12), 10685-10693.

[32] Tárrega, A., & Costell, E. (2006). Effect of inulin addition on rheological and sensory properties of fat-free starch-based dairy desserts. International Dairy Journal, 16(9), 1104-1112.

[33] Adhikari, B., Howes, T., Bhandari, B. R., & Truong, V. (2001). Stickiness in foods: a review of mechanisms and test methods. International Journal of Food Properties, 4(1), 1-33.

[34] Weenen, H., Van Gemert, L. J., Van Doorn, J. M., Dijksterhuis, G. B., & De Wijk, R. A. (2003). Texture and mouthfeel of semisolid foods: Commercial mayonnaises, dressings, custard desserts and warm sauces. Journal of Texture Studies, 34(2), 159-179.

Point 2. Add an explanation of the statistical analysis under each figure and table as I requested in review (explain what the letters abc etc mean).

Response 2: Thank you for your kind guidance. According to your comments, I added the meaning of “a, b, and c” in the part of 2.10 in the article as below:

“Notably, a, b, and c in the results represent statistically significant differences in data results.”

Reviewer 3 Report

The manuscript has been improved and can be accepted in the present form.

Author Response

(The authors gave the same response as above.)
